# Not Only Mania or Depression: Mixed States/Mixed Features in Paediatric Bipolar Disorders

**DOI:** 10.3390/brainsci11040434

**Published:** 2021-03-29

**Authors:** Delfina Janiri, Eliana Conte, Ilaria De Luca, Maria Velia Simone, Lorenzo Moccia, Alessio Simonetti, Marianna Mazza, Elisa Marconi, Laura Monti, Daniela Pia Rosaria Chieffo, Georgios Kotzalidis, Luigi Janiri, Gabriele Sani

**Affiliations:** 1Department of Neuroscience, Section of Psychiatry, Università Cattolica del Sacro Cuore, 00100 Rome, Italy; delfina.janiri@uniroma1.it (D.J.); eliana.conte01@icatt.it (E.C.); ilaria.deluca02@icatt.it (I.D.L.); mariavelia.simone01@icatt.it (M.V.S.); lorenzo.moccia@policlinicogemelli.it (L.M.); Alessio.Simonetti@bcm.edu (A.S.); marianna.mazza@policlinicogemelli.it (M.M.); elisa.marconi@guest.policlinicogemelli.it (E.M.); laura.monti@policlinicogemelli.it (L.M.); luigi.janiri@unicatt.it (L.J.); 2Department of Psychiatry and Neurology, Sapienza University of Rome, 00168 Rome, Italy; 3Eating Disorders Treatment Unit, Casa di Cura Villa Armonia Nuova, 00100 Rome, Italy; 4Menninger Department of Psychiatry and Behavioral Sciences, Baylor College of Medicine, Houston, TX 77030, USA; 5Department of Psychiatry, Fondazione Policlinico Universitario Agostino Gemelli IRCCS, 00100 Rome, Italy; 6Clinical Psychology Unit, Fondazione Policlinico Gemelli IRCCS, 00168 Rome, Italy; danielapiarosaria.chieffo@policlinicogemelli.it; 7NESMOS Department, Faculty of Medicine and Psychology, Sapienza University of Rome, Rome, Italy; giorgio.kotzaidis@uniroma1.it

**Keywords:** mixed state, mixed features specifier, bipolar disorder, childhood, adolescence, prevalence

## Abstract

*Background:* early onset is frequent in Bipolar Disorders (BDs), and it is characterised by the occurrence of mixed states (or mixed features). In this systematic review, we aimed to confirm and extend these observations by providing the prevalence rates of mixed states/features and data on associated clinical, pharmacological and psychopathological features. *Methods:* following the Preferred Reporting Items for Systematic Reviews and Meta-Analyses (PRISMA) guidelines, we searched from inception to 9 February 2021 for all studies investigating mixed states/mixed features in paediatric BD. Data were independently extracted by multiple observers. The prevalence rates of mixed states/features for each study were calculated. *Results:* eleven studies were included in our review, involving a total patient population of 1365 individuals. Overall, of the patients with paediatric age BD, 55.2% had mixed states/features (95% CI 40.1–70.3). Children with mixed states/features presented with high rates of comorbidities, in particular, with Attention Deficit Hyperactivity Disorder (ADHD). Evidences regarding the psychopathology and treatment response of mixed states/features are currently insufficient. *Conclusions:* our findings suggested that mixed states/features are extremely frequent in children and adolescents with BD and are characterised by high levels of comorbidity. Future investigations should focus on the relationship between mixed states/features and psychopathological dimensions as well as on the response to pharmacological treatment.

## 1. Introduction

Bipolar disorders (BDs) consist of manic, depressive, and mixed episodes, which may be separated by periods of euthymia. Mixed episodes were considered those episodes meeting criteria for *both* mania and depression in the DSM-III to DSM-IV-TR editions, while the DSM-5 introduced the “with mixed features” specifier, which could apply to any type of episode of BD and major depressive disorder (MDD). Currently, in the DSM-5, for manic/hypomanic episodes to meet criteria for mixed-features specifier, at least three depressive symptoms should be present nearly every day whereas, for a depressive episode to meet the same specifier, three manic/hypomanic symptoms should be present during the majority of days. Mixed states were described for the first time in ancient Greece and the Roman Empire (Aretæus of Cappadocia, 2nd Century (Ἀρεταῖος)), but not related to BD until 1899, when Emil Kraepelin (1856–1926) and his disciple Wilhelm Christian Jakob Karl Weygandt [1,2] described with extreme clarity the disease entity of manic–depressive illness. Thereafter, the first edition of the DSM classified cases with marked mixtures of the cardinal manifestations of mania and depression (mixed type). The subsequent editions applied the term “mixed” to similar concepts. However, a specific interest for mixed states appeared in the medical literature in the 1990s, and this was by Italian investigators [3,4,5] and one Armenian-American author [6]. Koukopoulos and his group, in particular, further pushed the mixed state to include agitated depression, at odds with the DSM-IV [7] distancing themselves from the DSM-5 [8], and contributed to fine-tuning the current concept [9]. Subsequently, other studies demonstrated that the clinical picture of mixed states is associated with a more severe form of BD. According to a recent review, patients with a combination of components of manic and depressive state tend to experience more mood episodes and more functional impairment [10]. In addition, they are more likely to relapse, to have lower inter-episode intervals, to experience more rapid cycling, and to have higher rates of suicide and comorbid conditions, such as substance abuse [10]. Furthermore, it is recognized that mixed manic/depressive presentations in BD predicted poorer pharmacological response [11].

The onset of BD is generally from 15 to 35 years of age, but earlier onset is not infrequent. In older years, onset was held to be mostly in adult ages, but it has recently become clear that many more cases have their onset prior to the end of adolescence [12]. Accordingly, for BD-I patients, onset occurred at an age lower than 12 years of age in 5% of cases, between 13 and 17 years of age (adolescence) in 28% of cases, while 53% occurred during the age range between 15 and 25 years, when the onset of BD peaks [13]. Cases with adolescent and adult onset do not differ for many measures and outcomes, but childhood onset cases developed more episodes per year, had more family history of BD, and displayed more psychiatric comorbidity than individuals with a later onset [13]. Furthermore, pooling together childhood and adolescent onset BD cases and compared to adult-onset patients, the former passed longer times with symptoms, displayed more polarity shifts, and showed more mixed symptoms [14]. With respect to adults with BD, in fact, children and adolescents are more likely to present with rapid cycling or to be in a mixed state [15]. Mixed states (or mixed features specifier) are a common occurrence in children and adolescents, not only in those diagnosed as having BD but also in those with a depressive episode. About 65% of a large sample with depression, either due to MDD or BD, met mixed state criteria, compared to only about 35% of children and adolescents who had only depression [16]. In one study, about 40% of adolescents with depression had BD and, of these, 82% exhibited mixed states [17], prompting the authors to conclude that mixed states were the most common presentation in this age. These observations are in line with the clinical practice among childhood and adolescence psychiatrists but still await confirmation. We are not aware of a systematic review of mixed states literature in paediatric age, including data on associated clinical and psychopathological features and response to pharmacological treatment. This study aims to fill this gap by providing prevalence rates, comorbidity, and clinical presentations of mixed states/features in childhood and adolescence in order to inform clinicians of the available evidence regarding intervention strategies.

## 2. Materials and Methods

### 2.1. Search Strategy

To investigate the epidemiology, features, and treatment response of mixed states/features in children and adolescents with BD, we conducted a PubMed search on 9 February 2021 using the following strategy: “mixed state *” AND (bipolar OR mood OR affect *) with the following filters applied: Humans, Child: birth–18 years, Newborn: birth–1 month, Infant: birth–23 months, Infant: 1–23 months, Child: 6–12 years, Adolescent: 13–18 years, Preschool Child: 2–5 years.

### 2.2. Eligibility and Study Selection

Inclusion criteria included original studies in peer reviewed journals focusing on alive children and/or adolescents with BD and reporting data on mixed states/features as defined by various versions of the American Psychiatric Association (APA) DSM, beginning from the third edition. Eligible studies had to report original data on patients with mixed states regarding their prevalence, clinical course or psychopathology, or their response to treatment. Both longitudinal and cross-sectional studies were admitted and could be retrospective or prospective. There were no time limits or language limits in regard to the selection of appropriate studies.

Exclusion criteria: Studies were excluded if they repeated the same data (characterised as “same as other”); if they were conducted on adult patients (appearing despite filtering on PubMed and marked as “Adults” on the Preferred Reporting Items for Systematic Reviews and Meta-Analyses (PRISMA) flowchart); if they used both children and adult populations without specifying which symptoms referred to which population or if they pooled different diagnoses without specifically reporting mixed states in patients with BD (marked as “Lumping” on the PRISMA flowchart); if the study did not focus on (marked as “Unfocused” on the PRISMA flowchart) or was unrelated to the subject matter (to mark as “Unrelated” on the PRISMA flowchart); if the studies did not report data on mixed states (marked as “No mixed” on the PRISMA flowchart); or if they were animal studies (despite filtering, to mark as “Animal” on the PRISMA flowchart), surveys of clinicians about patient symptoms without an actual patient population (marked as “Survey” on the PRISMA flowchart), post mortem studies (marked as “Postmortem” on the PRISMA flowchart), case reports or series (marked as “Case” on the PRISMA flowchart), opinion papers such as editorials and letters to the editor (with no data, hypotheses, or observations on other published studies) (marked as “Opinion” on the PRISMA flowchart), and meta-analyses and reviews (marked as “Review” on the PRISMA flowchart). The latter were used to search among their references for possible further eligible studies.

All authors participated in the selection of eligible studies to include through Delphi rounds. Two rounds were sufficient to reach complete agreement. We adhered to the *P*referred *R*eporting *I*tems for *S*ystematic Reviews and *M*eta-*A*nalyses (PRISMA) Statement [18].

### 2.3. Data Extraction and Synthesis

The data of selected papers were extracted using a standardised spreadsheet. All authors introduced relevant data in the summary table and mutual comparison ensued on deciding which features to retain. The numbers of individuals with mixed features/mixed states were extracted from each study in order to provide prevalence rates, expressed as percentages.

## 3. Results

Our search PubMed strategy on the aforementioned date produced 96 records, which was added to another 5, obtained through alternative strategies, amounting to a grand total of 101 articles to assess. Eligible articles spanned from 1995 to 2015. The inclusion process is highlighted in the PRISMA flowchart in Figure 1, which reports the reasons for exclusion. A summary of the included articles is displayed in Table 1. Of the included studies, five were longitudinal and six were cross-sectional. Three studies were interventional, and the others were observational. The eleven included studies involved a patient population of 1365 individuals, of whom 558 (40.88%) were female and 807 (59.12%) were male. Their age ranged from 3.5 to 20 years (we included two studies that comprised a few young adults with BD who had their onset during adolescence [19,20]). The prevalence rates of mixed features/mixed states in each study are reported in Figure 2. Overall, of the patients with paediatric age BD participating in eligible studies, 55.2% had mixed states/features (95% confidence intervals from 40.1 to 70.3). The clinical and psychopathological characteristics, and pharmacological treatment are reported in Table 1.

Comorbidities in the entire sample were variable and dependent on the authors’ main focuses. In the first eligible study [25], comorbidity with substance use was observed in 22% of cases during the 5-year follow-up. In another cross-sectional study [26], comorbidity rates were high, although not specified in patients showing mixed symptoms but in the entire sample without ADHD (oppositional defiant disorder, 88%; past major depressive episodes, 86%; at least two anxiety disorders, 44%; conduct disorder, 37%; and tic disorders, 26%). Similar high-comorbidity results were reported by another study, i.e., oppositional defiant disorder, 100%; attention deficit/hyperactivity disorder (ADHD), 78%; phobia, 57%; separation anxiety disorder, 52%; overanxious disorder, 48%; conduct disorder, 35%; and obsessive-compulsive disorder, 35% [22], and still another of the same time period, which reported comorbidities approaching 100% of BD with ADHD and with 70% comorbidity with any anxiety disorder [29]. Another study, relying on symptoms appearing in medical records, reported in their BD sample 42% ADHD, 30% schizophrenia, 25% conduct disorder, and 18% PTSD [19]. However, BD–schizophrenia cooccurrence is unlikely and the results of this study must be interpreted with caution. Another study focused on only BD–ADHD comorbidity, reporting it to be 54.6%, with 24% of patients making use of stimulant medications [23], while another that examined multiple comorbidities found ADHD to cooccur with ADHD in 60% of cases, of which 13% had cooccurrence of ADHD, depression, and anxiety [20]. Other studies reported no comorbidity at all [15,21,24,28], although recognising BD-ADHD comorbidity to be an issue and to affect treatment strategies [21].

**Table 1 brainsci-11-00434-t001:** Summary of the 11 eligible studies included in this review in order of year of publication.

Study	Design/Clinical Assessment	Population	Results	Conclusions
Strober et al., 1995 [25], Los Angeles, CA	LS; 5-yr follow-up; clinical observation, RDC (a predecessor of the DSM-III with stricter criteria) snd SADS	*N* = 54 BD-I; 29 ♀, 25 ♂; Age, x = 16.0 years; range 13–17 years	On admission, 20 were manic, 14 were depressed, 10 were mixed, and 10 were cycling; manic or mixed presentation recovered in 9 or 11 weeks, respectively, whereas depressive recovered in a median time of 26 weeks and rapid cycling recovered in 15 weeks	About 19% of adolescents hospitalised for BD-I present with mixed symptoms; the presence of a depressive core is a factor delaying recovery during a 5-year observation period
Wozniak et al., 1995 [26], Boston, MA	CS; referrals assessed for symptoms; DSM-III-R	*N* = 262 referrals for outpatient psychopharmacological treatment, of whom 207 had either mania (*N* = 43, age x = 7.9 ± 2.6 years; 36 ♂, 7 ♀) or ADHD (*N* = 164, age x = 8.8± 2.2 years; 141 ♂, 23 ♀); compared with 84 non-ADHD controls (age x = 9.0± 2.1 years; 71 ♂, 13 ♀); age, <12	Of children with mania, 14% (*N* = 6) had mania only, 84% (*N* = 36) had a mixed presentation, and 2% (*N* = 1) had nonoverlapping episodes of mania and major depression	Most children below age 12 present with mixed episodes (84%)
Biederman et al., 2000 [21], Boston, MA	LS, RChR; effect of treatments; DSM-III-R, K-SADS-E	*N* = 59 referrals for outpatient psychopharmacological treatment with BD (49 ♂, 10 ♀; Age, *x =* 10.8 ± 3.7 years; range 3.5–17 years)	The occurrence of mixed episodes was frequent in the sample. The patients received TCAs (45%), stimulants (20%), SSRIs (20%), mood stabilisers (40%), or FGAs (10%) during their follow-up visits. Depression was more responsive to SSRIs than to other treatments	This study supports the use of antidepressant monotherapy in youths with mixed states. The study was conducted during 1991–1995; at those times, SSRIs were not completely introduced in the US market. This justifies the lower rate of SSRI prescription compared to TCAs.
Frazier et al., 2001 [22], Boston, MA	LS; 20 mg/d open-label olanzapine × 8 weeks at 1 site; DSM-IV; assessment through the CGI-S and the YMRS	23 BD-I and -II (manic, mixed, or hypomanic; age, *x =* 10.3 ± 2.9 years, range 5–14 years; 13 ♂, 10 ♀)	Improvement criteria: at least 30% drop of YMRS score from baseline levels plus maximum 3 on the CGI-S mania, 61% responder rate; 17 patients (74%) with mixed presentation	No reported differential response for patients with mixed symptoms
DelBello et al., 2002 [15], Cincinnati, OH	LS; add-on quetiapine vs. placebo to valproate; random allocation × 6 weeks, double-blind, parallel; DSM-IV; YMRS primary outcome	30 BD-I patients with manic or mixed episodes randomised to valproate + quetiapine (*N* = 15; age, *x =* 14.1 ± 2 years; 8 ♂, 7 ♀) or valproate + placebo (*N* = 15; age, *x =* 14.5 years ± 2; 8 ♂, 7 ♀)	13 (87%) of patients randomised to valproate + placebo and 10 (67%) to valproate + quetiapine. Valproate + quetiapine reduced YMRS significantly better than valproate + placebo (but 7 patients in the former group quit the study before endpoint)	High attrition rate in the valproate + quetiapine group was not addressed through LOCF. No differential data provided for mixed episode BD-I patients. Small sample size
Wilens et al., 2003 [27], Boston, MA	CS, RChR; included if meeting DSM-III-R criteria. Assessed with K-SADS-E preschoolers 4-6-yr-old referred to a psychopharmacology clinic for BD and BD probands 7–9-yr-old	44 preschoolers with BD (age, *x =* 5.1 ± 0.8 years; 35 ♂, 9 ♀) 29 BD probands (age, *x =* 7.8 ± 0.9 years; 24 ♂, 5 ♀)	The two samples did not differ for psychopathology, other than conduct disorders, largely explained by oppositive-defiant disorder; 80% of preschoolers and 76% of school-age probands had mixed episodes	Children 4–6-year-old and those of 7–9 years of age with BD share psychopathological features and report similar rates of mixed states
Jerrell & Shugart, 2004 [19], USA	CS, RChR; utilised a cross-national database of clinical records of children with various psychopathological disorders diagnosed with the DSM-IV	Identified 83 patients with BD (age, *x =* 15 years; 43 ♂, 40 ♀); 16 were aged 7–12 years, 67 were 13–20-years-old	Most patients shared ADHD and conduct disorder symptoms; the two age groups did not differ in symptoms, save for the fact that very early onset children were more distractible, excitable, irritable, and fidgety and that early onset BD children displayed more depression; 23% had mixed episodes	About one fourth of patients with preadolescent and adolescent paediatric BD have mixed symptoms; in both, high rates of ADHD and conduct disorder comorbidity
Dilsaver & Akiskal, 2005 [28], Merced, CA	CS; assessment with SCID-DSM-IV of adolescents referred for MDD (MDE present)	49 consecutive Hispanic adolescents (age range 23–27 years; 33 ♀ with age x = 14.8 ± 1.6 years, 16 ♂ with age x = 14.8 ± 1.8 years)	Most patients were rediagnosed; 55% BD, 8.2% BD-I, 6.1% BD-II, and 40.9% mixed state. Half of the ♀ sample had family history positive for mood disorder and about one fourth had family history positive for MDD; 17 (51.5%) ♀ and 10 (62.5%) ♂ met DSM-IV criteria for BD. Psychotic features were more present in MDD than BD girls; the opposite was true for boys; irritability was shared by about 80% of the sample and hostility by about 50%	Most adolescents have more mixed states than pure depressive or manic/hypomanic presentations during a MDE. More than 40% of Hispanic adolescents present with mixed episodes, in line with results of other adolescent populations
Dilsaver et al., 2005 [24], Merced, CA	CS; acreening adolescent Hispanic (99%) destitues for mixed states with the SCID-DSM-IV	247 adolescents with MDE screened for mixed states (age, x = 14.7 ± 1.5 years; range, 12–17 years; 108 ♂, 139 ♀)	100 patients were with BD (40.5%); of them, 82 had a mixed state (46 ♂, 36 ♀, 33.1% of the whole sample); 147 met MDD criteria; 99 displayed psychotic features; 164 had suicidal ideation, 118 had past suicide attempts; 101 had positive family history for mood disorders (57 MDD, 44 BD). Only 11 were purely depressive BD-I and 7 purely depressive with BD-II	Confirmed was the high occurrence of mixed states in the presentation of adolescent patients with BD; the nonmixed samples did not differ from the mixed one for suicidal ideation or suicidal attempts
Potter et al., 2009 [20], Boston, MA	CS, RChR; youths diagnosed with DSM-IV BD at a paediatric psychopharmacology clinic; assessed severity according to treatment received; responder was a patient with CGI-I 1 or 2	53 (age, x = 13 ± 3.6 years, range 4–19 years; 18 (34%) <12-year-old, 39 (74%) ♂, 14 (26%) ♀)	Treated for comorbidity, 32 (68%); for ADHD, 32 (60%); for depression, 18 (34%); and for anxiety, 14 (26%). Monotherapy in 23% of patients, mostly SGAs. Responder rates, 80% of mania, 57% of mixed states, 56% of ADHD, 61% of anxiety, and 90% of depression	Provided response for mixed state but pooled numbers of patients with mania with those with mixed state. Results show mania to be a more unstable state than the mixed and a more responsive to SGAs
Findling et al., 2015 [23], multicentre (76 US; 10 Russia)	LS; double-blind randomised 1:1:1:1 assignment to 5, 10, and 20 mg/day asenapine or placebo to patients with manic or mixed episodes (DSM-IV-R) × 3 weeks; assessment with K-SADS, YMRS, CGI-BP; responder: who dropped ≥50% on the YMRS from baseline	403 patients (age, x = 13.8 ± 2.0 212; range, 10–17 years; ♀ (52.6%)) with mania (*N* = 171) or mixed state (*N* = 232) randomised to placebo (*N* = 101), 5 mg (*N* = 104), 10 mg (*N* = 99), or 20 mg asenapine (*N* = 99)	ADHD comorbidity *N* = 220 (54.6%), total comorbidities, *N* = 256 (63.5%), with stimulant use *N* = 96 (23.8%); asenapine performed better than placebo (lower and higher dosages, but all doses were equally efficacious on the CGI-BP). Responder rates, 42–54% with asenapine, 28% with placebo	Mixed states still constitute the majority of BD presentations in children and adolescents; more than one fourth of patients are likely to respond to placebo at the 3-week endpoint. Site effects analysed but not reported; separate analyses carried out for ADHD comorbidity (with or without) but not for pure mania vs. mixed state responsiveness

*Abbreviations*. ADHD, attention deficit/hyperactivity disorder; BD, bipolar disorder; BD-I, type 1; BD-II, type 2; CGI-BP, Clinical Global Impressions—Bipolar disorder; CGI-I, Clinical Global Impressions—Improvement; CGI-S, Clinical Global Impressions—Severity; CS, cross-sectional study; FGAs, first-generation antipsychotic drugs; HCs, healthy controls; K-SADS-E, Kiddie Schedule for Affective Disorders and Schizophrenia for School-Age Children—Epidemiologic version; LOCF, last observation carried forward; LS, longitudinal study; pts, patients; MDD, major depressive disorder; MDE, major depressive episode; RChR, retrospective chart review; RDC, Research Diagnostic Criteria; SADS, Schedule for Affective Disorders and Schizophrenia for school-age children; SCID-DSM-IV/-5, Structured Clinical Interview for the DSM-IV/5; SD, standard deviation; SGAs, second-generation antipsychotic drugs; SSRIs, selective serotonin reuptake inhibitors; TCAs, tricyclic antidepressants; x, mean; YMRS, Young’s Mania Rating Scale; ±, SD; ♀, female; ♂, male.

## 4. Discussion

In this systematic review, we confirmed what has always been a common impression among clinicians, i.e., that mixed states are frequent in children and adolescents. We found, in fact, that 55.2% of young individuals with BD presented with mixed states/features. This means that at least half of paediatric patients with BD experienced symptoms of the opposite polarity during mania/depression.

Why are mixed states/features so common in children and adolescents? One factor that may underpin this high prevalence could be emotional dysregulation. Emotion dysregulation is defined by difficulties in monitoring and evaluating emotional experiences, in modulating their intensity or duration, and in adaptively managing emotional reactions [30]. High levels of emotional dysregulation have been reported in adolescence [31] and may be explained by neurobiological substrates. It should be stated, in fact, that children and adolescents are in a condition of continuous brain maturation, which contributes to their being emotionally labile [32,33]., Emotional lability may depend on changes in subcortical structures, particularly in the limbic system [34], which has been reported as biomarkers of BD [35]. Specifically, emotional lability is a feature of BD and it has also been highlighted as a core feature of mixed states [36,37]. Interestingly, recent investigations found that the effect of BD diagnosis on subcortical structures, in particular limbic areas involved in emotion regulation, was mediated by traumatic events occurring during childhood [38,39]. Accordingly, a recent study hypothesised that childhood trauma could be related to the development of mixed states thought the mediation effect of emotional dysregulation /emotional lability [40].

In terms of clinical presentation, in this study, we found in children with mixed states/features high rates of comorbidities with other psychiatric disorders typical of the paediatric age. In particular, the largest agreement among the included studies was in reporting high rates of attention deficit hyperactivity disorder (ADHD), oppositional defiant disorder, and anxiety disorders. Anxiety disorders and BD are often comorbid and share genetic risk factors [41]. In particular, a recent study found that comorbid anxiety disorders reflect a dual burden of BD and anxiety-related genes [41]. Furthermore, anxiety disorder comorbidity adversely impacts the clinical course of the disease [42] and a recent meta-analysis confirmed that they are highly comorbid not only with adult but also with paediatric BD [43]. Oppositional defiant disorder and conduct disorder are very typically diagnoses of children and adolescents. They refer not only to the mood alterations that define BD but also to the behavioural alterations that may be encountered in social interactions. Nevertheless, it is well recognized that oppositional defiant disorder and BD and are highly comorbid. In particular, oppositional defiant disorder comorbidity has been predominantly demonstrated in patients with cooccurring BD and ADHD [44]. ADHD is also often encountered in comorbidity with BD in the adult age, although adult ADHD is often overlooked by adult psychiatrists when other explanations of the patient’s clinical status are available. Comorbid ADHD is present in up to 58.6% of paediatric patients with BD [45], while in children and adolescents with ADHD, the cooccurrence with bipolar disorder is 22% [46]. Interestingly, emotional dysregulation/lability is a core symptom of both paediatric and adult ADHD [47] and a predictor of adult emotional lability [48]. Therefore, it is possible that similar neurodevelopmental alterations result in abnormal neuronal firing in both BD and ADHD, which both display emotional lability. This hypothesis supports the more general concept that psychiatric disorders could share common neurobiological substrates, including neural activity [49] and connectivity patterns [50].

Available evidences regarding the treatment of mixed states/features in BD are currently insufficient to draw conclusions in our systematic review. Of the 11 eligible studies, five dealt with the effects of pharmacotherapy, but of these, two were observational, two were open label, and only one was double blind. The two observational studies assessed the effects of ongoing drug treatment, both considering a similar broad spectrum of drugs from mood stabilisers to antidepressants, stimulants, and antipsychotics [20,21]. Of the experimental studies, all used second-generation antipsychotics; the two open-label studies used 20 mg/day olanzapine [22] and 450 mg/day quetiapine as add-on to valproate [15], while the only double-blind study used 20 mg/day asenapine [23], the maximum dose for children. Only the latter was based on a relatively large sample. None of these drugs showed higher or lower efficacy in mixed states. It is surprising that, despite the high frequency of mixed states/features in children and adolescents, few studies have addressed the issue of their pharmacological treatment. One reason could be that the field of child and adolescent psychopharmacology is relatively new, and another is that there are fewer treatments approved for paediatric use than for adults. Another reason could be that psychiatrists in childhood, such as in the adult age [8], tend to primarily consider mania and depression [51] rather than focus on mixed features. Future studies should focus on mixed states and pharmacological treatment and, in particular, on treatment response to lithium, which is considered the goal standard among mood stabilizers and is associated with changes in subcortical structures [52].

A similar uncertainty is seen in psychopathological issues. Although the scientific community is increasingly endeavouring in mixed state/features matters, there is still insufficient focus on psychopathological features. None of the studies considered in this review addressed specific dimensions, such as irritability, agitation, anhedonia, and impulsiveness, which have extensively been described in mixed states [53]. Similarly, little attention has been paid to suicidal risk. It is important to rate mixed symptoms in paediatric patients with BD, since suicidal ideation and behaviours are not infrequent in childhood [29] and mixed features are associated with increased risk of suicide in BD [54]. However, in the only included study that focused on suicide ideation and attempts [24], no differences were reported in both measures between mixed and non-mixed symptom patients. Furthermore, an important issue is the possible diagnostic confusion between mixed features/states and rapid cycling. Although this point has been focused on in the various DSM versions, mixed states and rapid cycling showed potential clinical overlap [55]. This could be partly related to a common pattern of “mood instability”, which may involve both the coexistence of mania and depression and a pattern of frequent mood episodes. Accordingly, there is a need for further investigation in this particular field in order to demarcate features of mixed state and rapid cycling during psychiatric assessment in BD.

*Limitations*. Our review has the limitation of the small number of included studies and the small sample sizes that most of them employed. The methodologies were heterogeneous and sometimes unfocused, thus leaving out important questions, such as individual symptoms of mixed states/features. Another limitation that led us to exclude many studies was that studies tended to include patients with mixed or manic states without diversifying them, a pitfall carried on from adult psychiatry. Included studies used three different versions of the DSM, with DSM-III and DSM-IV radically differing in their views of mixed states from the DSM-5. This heterogeneity did not allow any meta-analysis of the results.

## 5. Conclusions

Summarising the evidence we obtained from the literature, mixed states are extremely frequent in children and adolescents. Nevertheless, methodologies in assessing them are heterogeneous and future studies are needed to corroborate diagnosis consistency. Furthermore, mixed states are characterised by high rates of comorbidity with typical paediatric psychiatric disorders, in particular, ADHD. The response to pharmacological treatment and the role of psychopathological dimensions have not been focused upon much in this specific literature, thus pointing to future investigational targets. In conclusion, mixed states need to be assessed in childhood and considered as a core feature of paediatric BD in direct opposition to the idea that mood disorders only swing from depression to mania.

## Figures and Tables

**Figure 1 brainsci-11-00434-f001:**
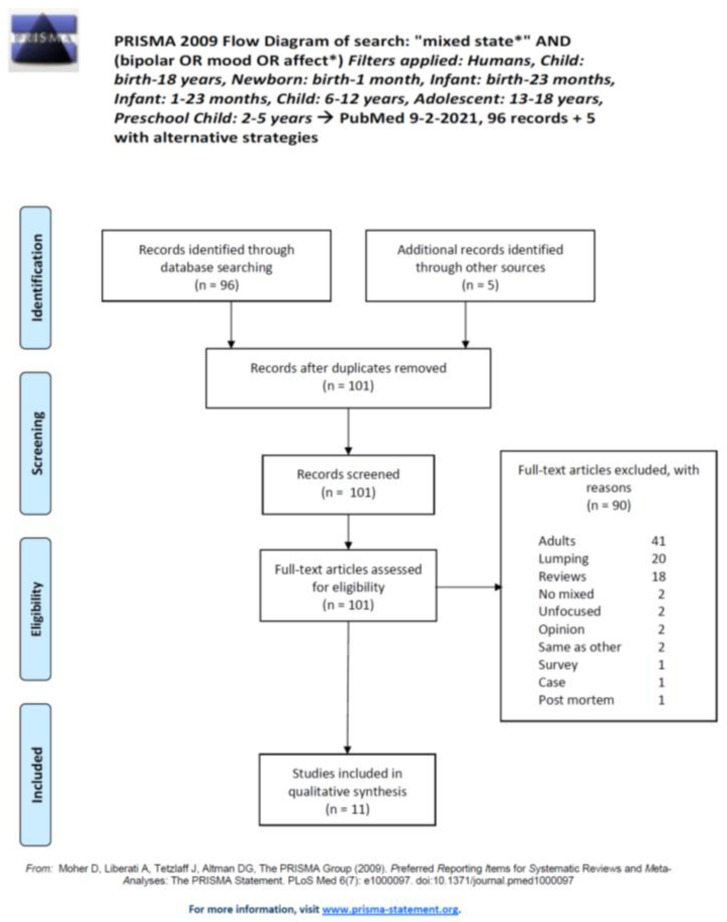
Preferred Reporting Items for Systematic Reviews and Meta-Analyses (PRISMA) flow diagram of the inclusion strategy with exclusion reasons (after Moher et al., 2009 [18]).

**Figure 2 brainsci-11-00434-f002:**
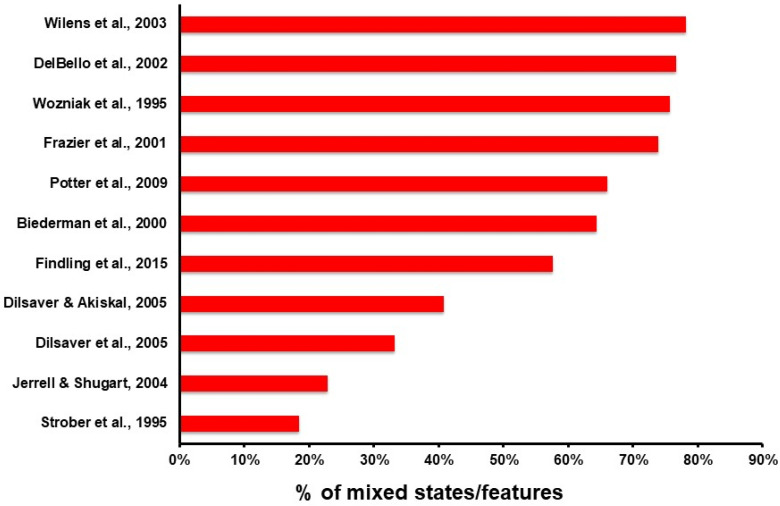
Prevalence rates of mixed features/mixed states in the included studies, from higher (top) to lower (bottom) [15,19,20,21,22,23,24,25,26,27,28].

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
