# Peer review of "Not Only Mania or Depression: Mixed States/Mixed Features in Paediatric Bipolar Disorders"

_brainsci, 2021, doi:10.3390/brainsci11040434_

Round 1

Reviewer 1 Report

In this systemic review article, authors reviewed 11 studies on mixed states/mixed features in pediatric BD. They reported that mixed states/features are frequent in pediatric BD population and showed increased comorbidities, especially for ADHD. The review was well written and very well organized. I only have few minor comments to improve the manuscript.

  • The authors can give a more detailed introduction on mixed states and add few examples about the prognosis.
  • In conclusions, authors report that children with mixed states/features presented with high rates of comorbidities, but they only discuss about ADHD. I would like a little bit discussion on other comorbidities as well.
  • The part about comorbidities only comes up in discussion and conclusions, but not in results sections.
  • Line 49, in “manic-depressive Illness” use lower case for “I”

Author Response

I held in this response the original observations of the reviewer and respond point-to-point underneath. We revised the paper according to your suggestions. Please find changes with respect to the previous version in red characters; deletions will not appear.

Comments and Suggestions for Authors

In this systemic review article, authors reviewed 11 studies on mixed states/mixed features in pediatric BD. They reported that mixed states/features are frequent in pediatric BD population and showed increased comorbidities, especially for ADHD. The review was well written and very well organized. I only have few minor comments to improve the manuscript.

 We thank the Reviewer for the positive attitude towards our paper.

  • The authors can give a more detailed introduction on mixed states and add few examples about the prognosis.

We thank the reviewer for the suggestion.  Accordingly, we expanded in the introduction section the description of mixed states in BD. In particular, we added more details of mixed state symptomatology and we better specified in the first part of the introduction the criteria for mixed features. Furthermore, we added a specific part focused on factors related to the clinical outcome.

  • In conclusions, authors report that children with mixed states/features presented with high rates of comorbidities, but they only discuss about ADHD. I would like a little bit discussion on other comorbidities as well.

We thank Reviewer for the suggestion. We added comorbidities ensuing from eligible studies in the result section and discussed them in the Discussion.

The part about comorbidities only comes up in discussion and conclusions, but not in results sections.

In the first version of the manuscript, we did not specify comorbidity rates because they were not always included in the original studies. We tried to overcome this by reporting comorbidity rates in each of the 11 eligible studies in the results section, specifying the studies that reported no comorbidity in the sample. Accordingly, we expanded the Discussion section.

  • Line 49, in “manic-depressive Illness” use lower case for “I”

We corrected, thank you, it was a misprint. Please find changes with respect to the previous version in red characters. We thank Reviewer for having carefully read our paper and for most appropriate suggestions that helped us to improve it.

Reviewer 2 Report

The authors presented a systematic review focused on the prevalence rates of mixed states/features in pediatric samples of Bipolar Disorder (BD). The review was prepared according to PRISMA guidelines and included eleven papers fulfilling the criteria for eligibility.
The analysis revealed a high rate of mixed state prevalence in the described clinical population what broadens the insight into the specificity of children/adolescents BD, however, some details should be addressed. 
- There are no specific data regarding assessment methods used by clinicians to set the diagnosis of BD with the mixed state. Was it made only on the basis of clinical impression (interview, observation etc.), or some more specific, validated methods were applied, for example with good psychometric properties? Table 1 should be should include an additional variable, such as "Methods of clinical assessment/diagnosis".
- Are the authors of the review sure that all researchers understood the very term "mixed state" the same? If not, it is difficult to aggregate data from different studies if the authors differed substantially in what the mixed states are and what symptoms must occur to make such a diagnosis.
- Does the included studies contained a very clear differentiation between mixed state and rapid cycling type of BD?
- Differential diagnosis between pediatric BD and ADHD should be analyzed and discussed in more details. Maybe some symptoms of affective instability and disinhibition occurring in ADHD were classified as symptoms od BD with mixed states?
- In general, a systematic review should not be so strictly focused on hypotheses confirmation, but more on objective analysis of included studies. A substantial prevalence of mixed states in the analyzed population should still be explained in a more critical way and the Authors should present various explanations of this state, also critical to the very results of the evaluated publications, which in fact contain methodological errors and are so heterogeneous in terms of diagnostic criteria that the review should be summarized with less "one-sided" evaluation.

Author Response

We revised the manuscript according to your suggestions. Changes will appear in red, while deletions will not appear. We kept your points here in this response and report underneath each point.

Comments and Suggestions for Authors

The authors presented a systematic review focused on the prevalence rates of mixed states/features in pediatric samples of Bipolar Disorder (BD). The review was prepared according to PRISMA guidelines and included eleven papers fulfilling the criteria for eligibility.
The analysis revealed a high rate of mixed state prevalence in the described clinical population what broadens the insight into the specificity of children/adolescents BD, however, some details should be addressed. 

We thank Reviewer for having positively considered our paper.

- There are no specific data regarding assessment methods used by clinicians to set the diagnosis of BD with the mixed state. Was it made only on the basis of clinical impression (interview, observation etc.), or some more specific, validated methods were applied, for example with good psychometric properties? Table 1 should be should include an additional variable, such as "Methods of clinical assessment/diagnosis".

We thank you for providing this useful suggestion. We added missing methods in the Design column of Table 1 to meet your requirement.

- Are the authors of the review sure that all researchers understood the very term "mixed state" the same? If not, it is difficult to aggregate data from different studies if the authors differed substantially in what the mixed states are and what symptoms must occur to make such a diagnosis.

We thank the reviewer for his/her observation. We totally agree with the reviewer on methodological concerns regarding heterogeneity in assessing mixed states. Following the reviewer’s suggestion, we added in the Design column of Table 1 information on the clinical assessment for mixed states diagnoses. According to the results, we found that most of the study consistently used similar diagnostic systems. Most involved authors, indeed, used the criteria of the various DSM-(III to 5) versions. Even though diagnostic criteria in the DSM versions varied over time, many of the changes in DSM-5 were made to better characterize symptoms and are based on a careful consideration of the scientific advances in research underlying the disorders. Accordingly, one of the goals of the DSM task force has been preserve diagnosis consistency. Please note that one statement in the Limitations expresses our worry that there might have been heterogeneous approaches in diagnosing mixed states.

- Does the included studies contained a very clear differentiation between mixed state and rapid cycling type of BD?

We thank the Reviewer for his thoughtful observation. Even though mixed states and rapid cycling share some common clinical characteristics, including high incidence of suicidal thoughts, high levels of anxiety, and previous substance misuse, most of the included studies adhered to DSM criteria in differentiating between the two. The clinical overlap between mixed states and rapid cycling could be partly related to a common pattern of “mood instability”, which may involve both the co-existence of mania and depression and a pattern of frequent mood episodes. Please note that we added a statement and a reference in the discussion section to specifically address this important point.

  • Differential diagnosis between pediatric BD and ADHD should be analyzed and discussed in more details. Maybe some symptoms of affective instability and disinhibition occurring in ADHD were classified as symptoms od BD with mixed states?

Please note that the comorbidity between ADHD and BD has been largely discussed in the Discussion section, with a specific focus on their shared clinical presentation. In particular, we focused on emotional dysregulation/lability, which is a core symptom of both BD and ADHD. We also hypothesized that similar neurodevelopmental alterations result in abnormal neuronal firing in both BD and ADHD, ultimately leading to a similar clinical presentation. Following the reviewer’s suggestion, we also added in the Discussion a specific part focusing on other BD comorbidities, in particular anxiety disorders and oppositional defiant disorder, as now reported in the result section.

  • In general, a systematic review should not be so strictly focused on hypotheses confirmation, but more on objective analysis of included studies. A substantial prevalence of mixed states in the analyzed population should still be explained in a more critical way and the Authors should present various explanations of this state, also critical to the very results of the evaluated publications, which in fact contain methodological errors and are so heterogeneous in terms of diagnostic criteria that the review should be summarized with less "one-sided" evaluation.

We thank the reviewer for raising this important issue. As we stated in the introduction, our systematic review aimed at describing prevalence rates and clinical presentations of mixed states/features in childhood and adolescence, in order to inform clinicians of the available evidence regarding intervention strategies, rather than testing an aprioristic hypothesis. We agree with the reviewer that selected studies were heterogeneous when assessing mixed stated and that this may represent a potential diagnostic pitfall. Nevertheless, most of the selected studies applied DSM informed criteria to ensure diagnostic consistency. However, in line with reviewer suggestions, we further expanded this issue by adding in the Design column of Table 1 information on the clinical assessment for mixed states diagnoses and by highlightening in the limitations section of the manuscript that methodologies in assessing mixed states lack homogeneity. Furthermore, we also acknowledged in the limitations section that studies tended to include patients with mixed or manic states without diversifying them, so we excluded most of them. Moreover, in the conclusion section, we added a specific statement referring to the need of future studies in corroborating the consistency of mixed states diagnosis.

We thank Reviewer for his/her suggestions that helped us improving significantly our manuscript.

Round 2

Reviewer 2 Report

The authors responded to all my suggestions.